# Palmitic Acid Upregulates CD96 Expression to Mediate Maternal–Foetal Interface Immune Tolerance by Inhibiting Cytotoxic Activity and Promoting Adhesion Function in Human Decidual Natural Killer Cells

**DOI:** 10.3390/bioengineering10091008

**Published:** 2023-08-25

**Authors:** Yingjie Wang, Yun Wang

**Affiliations:** Department of Assisted Reproduction, School of Medicine, Shanghai Ninth People’s Hospital Affiliated to Shanghai Jiao Tong University, No. 500 Zhizaoju Road, Huangpu District, Shanghai 200025, China; wangyj_20@sjtu.edu.cn

**Keywords:** CD96, dNK cells, immune adhesion, maternal–foetal immunology, recurrent spontaneous abortion

## Abstract

Decidual natural killer cells (dNK cells) are an essential component of the immune cells present at the maternal–foetal interface during early pregnancy, and they play a vital role in various physiological processes. Abnormalities in the ratio or function of dNK cells have been linked to recurrent miscarriages. CD96 has been previously shown to regulate NK cell function in the tumour microenvironment; however, its role and mechanism at the maternal–foetal interface remains unclear. The present study aimed to investigate the immunomodulatory role of CD96 in dNK cells and its function at the maternal–foetal interface. Immunofluorescence staining and flow cytometry were used to detect the expression of cellular markers such as CD96. Furthermore, the secretory function, adhesion-function-related molecules, and cell proliferation markers of CD96+ and CD96− dNK cells were detected using flow cytometry. In addition, we performed cell culture experiments via the magnetic bead sorting of NK cells to detect changes in the expression of the aforementioned functional molecules in dNK cells after the CD96 blockade. Furthermore, we examined the functional characteristics of dNK cells after palmitic acid treatment at a concentration of 10 μM. We also examined the changes in dNK cell function when subjected to the combined effect of palmitic acid and CD96 antagonists. The results indicated that CD96, TIGIT, CD155, and CD112 were highly expressed at the maternal–foetal interface, with dNK cells predominantly expressing CD96, whereas TIGIT was mainly expressed on T cells, and CD155 and CD112 were mainly present in metaphase stromal and trophoblast cells. CD96+ dNK cells displayed low cytotoxic activity and a high adhesion phenotype, which mediated the immunosuppressive effect on dNK cells at the maternal–foetal interface. Palmitic acid upregulated CD96 expression on the surface of dNK cells in the coculture system, inhibiting dNK cell activity and increasing their adhesion molecule expression. CD96 antagonist treatment blocked the inhibitory effect of trophoblasts on dNK cells, resulting in enhanced cytokine secretion and reduced adhesion. The results of this study provide valuable insight into the immunomodulatory role of CD96 in dNK cells and its mechanism at the maternal–foetal interface, particularly in metaphase NK cells. This study sheds light on the mechanisms of immune regulation at the maternal–foetal interface and their implications for the study of recurrent miscarriages of unknown origin.

## 1. Introduction

The maternal–foetal interface is a key site for the establishment and maintenance of normal pregnancy, and it is mainly composed of trophoblast cells, decidual immune cells, and decidual stromal cells [1,2]. More than 30% of decidual cells in early pregnancy are immune cells, which are one of the main components of the maternal–foetal interface [3]. A variety of immune cells participate in the maintenance of maternal and foetal immune homeostasis during pregnancy, such as NK cells, T cells, macrophages, and dendritic cells [4]. The proportion and phenotype of these cells in the process of pregnancy change dynamically, indicating that the composition of immune cells is different during pregnancy, in different stages of pregnancy, and in different states of pregnancy [5,6]. It has been shown that immune homeostasis at the maternal–foetal interface is usually disrupted in patients with recurrent pregnancy failure [7].

Interactions with trophoblast cells through surface molecules or secreted substances affect the process of pregnancy and are important for tolerating and maintaining normal foetal growth [8]. Studies have found that a large number of dNK cells that appear at the maternal–foetal interface aggregate, but they do not only exert cytotoxic effects like peripheral NK cells. Moreover, there is growing evidence to suggest that the existence of dNK cells is also an important factor in maintaining pregnancy [9]. CD56, as a characteristic marker molecule of NK cells, exhibits a CD56dim phenotype on highly cytotoxic NK cells in peripheral blood, though it displays a CD56bright phenotype on decidual NK cells in the decidua, indicating a close association between CD56 expression and the functional activity of NK cells. Accordingly, the composition of immune cells and cytokines at the maternal–foetal interface maintains dynamic changes. As pregnancy progresses, the microenvironment during embryo implantation and placental development gradually changes from a proinflammatory phenotype to an anti-inflammatory phenotype; for example, dNK cells [10]. Williams et al. found that the number of decidual CD56+ cells did not change in the first and second trimester, and CD56+ cells were reduced in the third trimester [11]. Chen et al. found that the proportion of CD56+ dNK cells with perforin and granzymes in the decidua gradually decreased as pregnancy progressed, suggesting that dNK cells function in a more moderate manner during pregnancy [12].

CD96 is a member of the poliovirus receptor (PVR, CD155)-connexin family, which includes the T-cell Ig and ITIM domains (TIGIT) and CD226 [13]. CD96, TIGIT, and CD155 play an important role in regulating NK cell activity, which relies on a large number of receptor combinations to initiate effector functions [14]. PVR, functioning as a high-affinity ligand for CD96, directly binds to CD96 receptors and mediates the inhibitory effect on the cytotoxicity of NK and T cells. These effects have been widely validated in tumour tissue, but it is unclear whether CD96 exhibits inhibitory or stimulatory functions in decidual NK cells [15].

Palmitic acid (PA) is one of the most common 16-carbon long-chain fatty acids. It is rich in the human body, and it is one of the most important raw materials for cholesterol synthesis and the main component of bile acids [16]. Palmitic acid is not only a main component of the constituent cell membrane, but it also exhibits biological activity in the immune system [17]. Previous studies have shown that high concentrations of palmitic acid can activate the inflammatory response through multiple pathways [18]. The excessive accumulation of palmitic acid induces an inflammatory response at the maternal–foetal interface, leading to adverse pregnancy events such as eclampsia or miscarriage [19]. Another previous study demonstrated that a concentration of 10 μM palmitic acid upregulated the expression of CXCL12 and IL15 at the maternal–foetal interface, potentially contributing to the prevention of recurrent implantation failure [20]. We suspect that a low dose of palmitic acid has more benefits for pregnancy maintenance at this time, but no studies have yet confirmed this conjecture.

The aims of this study were to investigate the role and molecular mechanism of CD96 in resident dNK cells during pregnancy, to investigate the influencing factors on the pathogenesis of spontaneous abortion, and to explore the function of palmitic acid in this process.

## 2. Materials and Methods

### 2.1. Tissues

All tissues were collected with the permission of the ethics committee and the consent of patients in the Obstetrics and Gynecology Hospital of Fudan University. Endometrial tissue was collected from women of childbearing age (25–40 years) with a normal history of pregnancy who underwent hysterectomy or diagnostic curettage for benign causes unrelated to endometrial dysfunction. All samples were evaluated by a histopathologist to rule out endometrial pathology and identify the cycle phase, specifically the secretory phase. None of the subjects had received hormonal medication in the preceding 6 months prior to surgery. Normal pregnancy decidual tissue was obtained from women with selective termination of pregnancy in the normal first trimester or with nongenetic or nonendocrine factors (age, 21–35 years; gestational age, 7–9 weeks). The decidua of spontaneous abortion came from women who underwent evacuation at 6–10 weeks of gestation when the foetal heart stopped. All tissues were collected under sterile conditions, and 10% foetal bovine serum (FBS) in DMEM/F-12 (HyClone, Logan, UT, USA, SH30023.01B) was added within 30 min along with 10% foetal bovine serum (FBS; Gibco, Grand Island, NY, USA, 26140-079) for further isolation of ESCs, DSCs, and dNK cells. ESCs, DSCs, and dNK cells were isolated and cultured according to our previously established protocol [21].

### 2.2. Cell Culture

The DSCs or dNK cells used in the experiment were primary cells obtained through the following process. Firstly, the tissues mentioned in Section 2.1 were cut into pieces and digested using collagenase. The resulting cell suspension was then filtered through a strainer to remove any debris. Subsequently, the cell suspension was centrifuged using a density gradient with Percoll reagent. This centrifugation step allowed for the separation of cells based on their density. After centrifugation, the cells were collected and cultured in well plates for further cell culture studies. The cell processing in this paper includes treatment with the CD96 antibody (Ultra-LEAF™ Purified anti-human CD96; 338422, Biolegend, San Diego, CA, USA) at a concentration of 10 µg/mL or palmitic acid (PA; SYSJ-KJ003/KC003, Kunchuang Biotechnology, Xi’an, China) at a concentration of 10 μM for 24 h. Afterwards, the protein levels of specific molecules or the adhesion of stromal cells to NK cells were detected using flow cytometry staining. Cytokines were detected for subsequent intracellular flow cytometry staining assays.

### 2.3. Lysis of Erythrocytes

If necessary, the sample was diluted 10× with deionized water, red blood cell (RBC) lysis buffer (BioLegend, San Diego, CA, USA, 420301) was diluted to a 1× working concentration, and then the precipitate was resuspended in 3 mL of 1× RBC lysis buffer. The cells were incubated on ice for 5 min. Cell lysis was stopped by adding 10 mL of cell staining buffer to the tube. The mixture was centrifuged for 5 min at 350× *g*, and the supernatant was discarded. The wash was repeated as described above.

### 2.4. Flow Cytometry Assays

Human antibodies for flow cytometry assays were used for the measurement of cell markers. Matched immunoglobulin G (IgG) antibodies were used as isotype controls. Flow cytometry assays were performed according to the manufacturer’s instructions. For the detection of intracellular molecules such as cytokines, we used Cell Activation Cocktail (with Brefeldin A) (Biolegend, San Diego, CA, USA, 423303) to stimulate the cells for 6 h prior to flow cytometry antibody staining of the cell surface. Fixation Buffer (Biolegend, San Diego, CA, USA, 420801) was used for cell fixation and Intracellular Staining Perm Wash Buffer (Biolegend, San Diego, CA, USA, 421002) was used for permeabilization followed by intracellular staining. The cells were subsequently stained with flow cytometry antibodies and detected on the machine. Cell sorting was performed with the NK cell magnetic bead sorting kit (Miltenyi Biotec, Shanghai, China, 130-092-657). The flow cytometry antibodies used are listed in Table 1. Data were analysed using FlowJo V10 software.

### 2.5. Paraffin Section Preparation

Fresh tissue was fixed with a fixative for more than 24 h. The tissue was removed from the fixation fluid in the ventilator, smoothed with a scalpel, and placed with the corresponding label in the dehydration box, which was placed in the dehydrator with alcohol gradients for dehydration: 75% alcohol for 4 h, 85% alcohol for 2 h, 90% alcohol for 2 h, 95% alcohol for 1 h, absolute ethanol I for 30 min, absolute ethanol II for 30 min, alcohol benzene for 5–10 min, xylene I for 5–10 min, xylene II for 5–10 min, 65 °C melted paraffin for 1 h, 65 °C melted paraffin II for 1 h, and 65 °C melted paraffin III for 1 h. Once the dehydration process was completed, the wax-soaked tissue was embedded in the embedding machine. The melted wax was first placed into the embedded box. Before the wax solidified, the tissue was removed from the dehydration box and placed into the embedded box according to the embedded face, and the corresponding label was attached. The frozen table was cooled to −20 °C, and the wax block was solidified, removed from the embedded box, and trimmed. The finished wax block was placed in a −20 °C frozen table to cool, and then the cooled wax block was placed in a paraffin-slicing microtome and cut to 4 μm thick. Floating slices were placed in 40 °C warm water to flatten the tissue, and the tissue was removed and baked in a 60 °C oven. After baking with water and dry wax, the sample was removed and kept at room temperature.

### 2.6. Immunofluorescence in Paraffin Sections

The paraffin wax-to-water procedure involved several sequential steps. Firstly, environmentally friendly dewaxing was performed twice for 10 min each time, followed by successive immersions in absolute ethanol for 10 min, 15 min, and 5 min, and finally in distilled water for 5 min. For antigen repair, tissue sections were placed in a repair box containing EDTA antigen repair buffer (pH 8.0) and subjected to microwave heating. The samples were initially heated at medium power until boiling for 8 min, followed by an 8 min cooling period, and finally heating at low power for 7 min. This heating cycle was designed to prevent excessive buffer evaporation while ensuring that the samples did not dry out. After natural cooling, the slides were washed in PBS (pH 7.4) for 5 min each. To facilitate serum closure, a circle was drawn around the tissue on the slightly dried section to prevent antibody flow. The sample was dried using PBS, followed by the addition of BSA. The sample was then sealed for 30 min, using 10% donkey serum for blocking antibodies from goat sources and 3% BSA for blocking antibodies from other sources. Next, the blocking solution was gently removed, and PBS was added to the section in a specific proportion. The slides were then placed flat in a wet box and incubated overnight at 4 °C. To prevent antibody evaporation, a small amount of water was added to the wet box. Following the primary antibody incubation, the slides were washed three times in PBS (pH 7.4) using a counter colour shaker for 5 min each time. After slight drying, the corresponding species-specific secondary antibody was added to the circle, and the sections were incubated for 50 min at room temperature. For DAPI counter-staining of the nuclei, the slides were washed three times in PBS (pH 7.4) on a colour shaker for 5 min each time. The sections were then slightly dried, and DAPI dye solution was added to the circle, followed by a 10 min incubation at room temperature. To quench spontaneous fluorescence of the tissue, the slides underwent three washes in PBS (pH 7.4) for 5 min each time. A spontaneous fluorescent quencher was added to the circle for 5 min, and the slides were subsequently washed with running water for 10 min. After slight drying, the sections were sealed with an antifluorescence quenching agent. Finally, the sections were examined under a fluorescence microscope, and images were collected using specific excitation and emission wavelengths: DAPI UV excitation at 330–380 nm and emission at 420 nm (blue light), FITC excitation at 465–495 nm and emission at 515–555 nm (green light), and CY3 excitation at 510–560 nm and emission at 590 nm (red light). The immunofluorescence results of the paraffin sections showed blue fluorescence of cell nuclei stained with DAPI under UV excitation, along with the corresponding red or green fluorescence from the fluorescein-labelled antibodies.

### 2.7. Cell Adhesion Assays

Stromal cells after different treatments of lentivirus infection were collected, washed, and cultured in a 24-well plate (1 × 10^5^ cells per well) with or without staining with PKH67 (Sigma-Aldrich, St. Louis, MO, USA, PKH67GL) overnight. PKH26-labelled (Sigma Aldrich, PKH26GL) dNK cells were cocultured with HTR8s or DSCs for 24 h. Then, for removal of unattached dNK cells, the medium was removed and the cells were washed gently with PBS twice. Pictures were obtained under a fluorescence microscope (Olympus, Tokyo, Japan, IXplore Standard) at five random fields of view and are shown as the ratio of dNK cells to the stromal cell layer [21].

### 2.8. Statistical Analysis

All experiments were repeated at least in triplicate. All data were analysed using GraphPad Prism version 8. All parameters were analysed using an unpaired *t* test, Mann-Whitney U test, or one-way ANOVA. Data that were normally distributed are presented as the mean ± STD. Statistical significance is indicated by *p* < 0.05.

## 3. Results

### 3.1. CD96 Is Enriched in the Maternal–Foetal Interface during Pregnancy

To clarify CD96 and CD155 expression at the maternal–foetal interface, we performed immunofluorescence staining of the normal endometrium (NE) tissues and decidua tissues from normal pregnancy (NP) women and spontaneous abortion (SA) women. In the decidua of normal pregnancies, the expression of CD96 and CD155 at the maternal–foetal interface was significantly higher compared to spontaneous abortion and normal endometrium (Figure 1). The expression of CD155 in the decidua and villi of patients with spontaneous abortion was relatively lower compared to normal pregnancy.

### 3.2. CD96 Is Highly Expressed in Normal Gestational Decidual NK Cells, While There Is No Difference in the Expression of TIGIT in the Endometrium and Normal Pregnancy Decidual NK Cells

To detect the cell specificity of CD96 expression in decidual and endometrial tissues, we labelled CD45 for lymphocytes, CD3 for T cells, and CD56 for NK cells with a flow cytometry antibody, and differences in CD96 expression were compared between cell groups (Figure 2A–D). We found that the expression of CD96 in various groups of cells in the decidua of normal pregnancy was significantly higher than that of the endometrial tissue. We also assessed the CD96-associated TIGIT marker, which was found to be involved in the immunosuppression of CD155 on NK cells in tumour tissue. Within the same tissue, TIGIT expression was significantly higher on T cells and total lymphocytes than on NK cells, either in endometrial tissue or in decidual tissue of normal pregnancy. In contrast, there was no difference in TIGIT expression on NK cells or T cells when comparing endometrial and decidual tissues. Since CD155 and CD96 are a pair of interacting molecules, it is necessary to detect the expression of CD155 at the maternal–foetal interface, and we also measured the expression of CD112, the binding site of TIGIT, on decidual stromal cells to determine the expression of this family of molecules on the decidual surface. We labelled DSCs (decidual stromal cells) and ESCs (endometrial stromal cells) with Vimentin to detect CD155 and CD112 expression on the surface of DSCs and ESCs, as shown in Figure 2E–G. Flow cytometry showed a higher expression of CD155 (*p* < 0.05) and CD112 (*p* < 0.01) in decidual tissue in normal pregnancy compared to endometrial tissue.

### 3.3. Phenotype and Characteristics of CD96+ dNK Cells

Primary decidual cells were labelled via flow cytometry, and the dNK cells of CD96+ and CD96− were windowed separately to detect the functional receptors of dNK cells. We investigated whether CD96 expression on the surface of dNK cells affects their function and phenotype, similar to the role of CD96 in pNK cells, laying down the foundation for subsequent research. We used flow cytometry staining to label the characteristic molecules CD56 and CD3 of NK cells and gated NK cells, and then further identified the negative and positive groups of CD96 on the surface of the NK cells. The results showed that the proportion of NK cells in primary immune cells was approximately 57% (Figure 3A). Among NK cells, the proportion of CD96+ NK cells was significantly higher than that of CD96− NK cells (Figure 3B). Various types of adhesion molecules exist according to their structural characteristics. In addition to some adhesion molecules that have not yet been classified, they can also be grouped into families such as the integrin family, selectin family, immunoglobulin superfamily, cadherin, etc. Here, we selected three representative adhesion molecules for determination: ICAM1 (CD54), VCAM1 (CD106), and selectin SELE (CD62E) of the immunoglobulin superfamily. Previous studies have found that these molecules are abundantly expressed on the surface of dNK cells localized to the decidua during pregnancy, thereby mediating the adhesion of dNK cells at the maternal–foetal interface. In our examination, CD96+ and CD96− NK cells were separately analysed, revealing that three adhesion molecules were markedly elevated on CD96+ NK cells (Figure 3C,D). NK cells can express a variety of cellular molecules, such as IFN-γ, granzyme B, TNF-α, and perforin, and the expression of cytokines is positively correlated with the killing activity of NK cells, especially in inflammatory environments. To verify the relationship between CD96 and dNK cell cytokine function, we detected CD96+ and CD96− NK cells and found that cytokines were significantly reduced on CD96+ NK cells (Figure 3E,F).

### 3.4. After CD96 Antagonists Block NK Cells, the Adhesion of NK Cells to Stromal Cells and Trophoblasts Decreases

To detect the role of CD96 at the maternal–foetal interface, we used CD96 antibodies as antagonists and added them to the coculture system at a concentration of 10 μg/mL. The adherent and suspended cells were separately labelled using the Cell Adhesion Assay Kit. Here, we labelled adherent cells with green fluorescence, i.e., DSCs or HTR8 cells, and suspended cells, i.e., NK cells, with red fluorescence (Figure 4A–D). The results showed that the adhesion of NK cells to stromal cells decreased after the addition of antagonists. In addition, we collected NK cells for adhesion molecular assays using flow cytometry and found that the expression of the three representative adhesion molecules decreased after the addition of CD96 antagonists (Figure 4E). This finding suggests that CD96 mediates the adhesion function of NK cells in situ, helping to maintain normal pregnancy. Adhesion decreased after the addition of antagonists. In the context of the NK cell and HTR8 coculture, changes in NK cell activity after the addition of CD96 antagonists were detected. We used flow cytometry to detect three cytokines within NK cells. Among them, IFN-γ is a cytokine associated with cytotoxicity and Ki-67 is a proliferation marker that is positively correlated with NK cell activity. The IL-10 inhibitory receptor molecule was negatively correlated with NK cell activity. After coculture for 24 h, the cells were stimulated with the cell activation cocktail. The results of flow cytometry showed that after the addition of antagonists, the expression intensity of IFN-γ increased significantly, the expression intensity of IL-10 decreased significantly, and the expression intensity of Ki-67 increased significantly (Figure 4F).

### 3.5. Low-Dose Palmitic Acid Can Regulate CD96 Expression, Thereby Affecting Cellular Oxidative Function

We detected CD155 molecules on the surface of HTR8 cells and CD96 molecules on the surface of dNK cells via flow cytometry after the addition of palmitic acid to the culture system with single culture or coculture of dNK cells and HTR8 cells. To further investigate the beneficial effects of low-dose palmitic acid at the maternal–foetal interface, we treated the cells with a low concentration of 10 μM palmitic acid. The expression of surface CD155 molecules was significantly higher in the PA group when 10 μM palmitic acid was added to HTR8 cells compared to the control group of HTR8, where no additional reagents were added or other treatments were performed in addition to the conventional medium (Figure 5A). And when HTR8 cells were cocultured with dNK cells, the addition of palmitic acid in the PA group was not significantly different from the control group without palmitic acid in terms of the expression of CD155 on the surface of HTR8 (Figure 5B). The PA group of dNK cells with palmitic acid addition showed no difference in CD96 expression compared to the control group of dNK cells cultured alone. In contrast, when dNK cells were cocultured with HTR8 cells, the expression of CD96 on dNK cells was significantly higher in the PA group with palmitic acid addition compared with the control group without palmitic acid addition (Figure 5C,D). To further explore the effect of palmitic acid on NK cells, we added CD96 antagonists to the culture system to investigate whether palmitic acid can affect the function of NK cells through its interaction with CD96. We tested the oxidative function of NK cells under different conditions using a Reactive Oxygen Species (ROS) probe kit, and the results showed that the addition of palmitic acid reduced the ROS content of NK cells both in the PA group and in the PA-HTR8 group cocultured with HTR8, compared to the control group in which NK cells were cultured without any treatment, indicating that palmitic acid inhibited the oxidative function of NK cells. After the addition of CD96 antagonists, the ROS content of NK cells was partially restored (Figure 5E). In addition, we used a mitochondrial membrane potential kit to identify changes in the mitochondrial membrane potential of cells by detecting changes in the ratio of JC-1 polymers/monomers, thereby reflecting alterations in cell activity. In general, a higher mitochondrial membrane potential indicates higher cell activity, and a lower mitochondrial membrane potential indicates possible apoptosis, while the three experimental groups after the addition of palmitic acid showed no statistic difference. The results showed that the addition of palmitic acid could potentially inhibit the mitochondrial membrane potential of NK cells compared to the control group, in which NK cells were cultured alone. However, when coculturing dNK cells with HTR8 cells, the addition of palmitic acid and the CD96 antagonist did not yield significant differences in the detection of mitochondrial membrane potential (Figure 5F). We examined changes in the expression of functional molecules in NK cells after the addition of palmitic acid and/or CD96 antagonists, using dNK cells and HTR8 cocultures as a control group. We used flow cytometry to assess three adhesion molecules within NK cells. These adhesion molecules were negatively correlated with NK cell activity. After a total of 24 h of incubation, flow cytometry was performed. The results of flow cytometry showed that after the addition of palmitic acid, the three adhesion molecules were upregulated as expected, indicating that the adhesion function of NK cells increased and shifted to a more stable phenotype. When CD96 antagonists were added, all three adhesion molecules decreased, suggesting that CD96 antagonists relieved the effect of palmitic acid on the enhancement of NK cell adhesion function (Figure 5G,H). In addition, we examined the expression levels of molecules in NK cells under the above conditions, and found that the expression levels of proliferation marker Ki-67 and cytokine IFN-γ decreased significantly after the addition of palmitic acid, while the expression levels of the two increased significantly after the addition of CD96 antagonists, which was not different from the control group. The expression level of inhibitory cytokine IL-10 is the opposite of the former (Figure 5I,J).

## 4. Discussion

In this study, we found that the expression of CD96 in dNK cells in the uterine decidua was significantly different from that in uNK cells in the endometrium. The low expression of CD96 in spontaneous abortion patients indicates the enhanced activity of NK cells, which may be one of the possible factors contributing to pregnancy failure. On the other hand, there is a strong correlation between spontaneous abortion and increased production of proinflammatory factors, such as cytokines TNF-α and IFN-γ, and conversely, a reduction in the anti-inflammatory cytokine IL-10 is also associated with spontaneous abortion. The associated cytokines and immune status were also verified in this experiment, demonstrating the decrease in granzyme B and IFNγ caused by the immunosuppressive state mediated by CD96. Notably, a recent study by Habets et al. found increased expression of CD96 on PBMCs from recurrent pregnancy loss samples, in contrast to the current study that found reduced CD96 expression on dNK cells. Many phenotypic differences between peripheral blood immune cells and maternal–foetal interface immune cells have been reported, and the underlying mechanisms of the differential phenotypic changes in CD96 at the peripheral blood and maternal–foetal interfaces need further exploration [22].

Indeed, CD155, CD96, CD112, and TIGIT form a subfamily of associated IgSF receptors that make up the stimulus/inhibition network. Some scholars classify these receptors as members of the CD155 family, and multiple intricate interactions exist among them [23]. Especially in tumour tissues, the interaction of TIGIT with CD155 has been applied to clinical tumour treatment [24]. Therefore, in this chapter, we conducted further testing of this family of molecules using flow cytometry, and the results showed that the expression of CD96 on immune cells was consistent with the results of immunofluorescence, while there was no significant difference in the expression of TIGIT between the endometrium tissue and normal pregnancy decidua. This led us to consider that CD96 may exert an immune tolerance function at the maternal–foetal interface by binding to CD155. Correspondingly, we measured the expression of CD155 and CD112 in the endometrium and decidua of normal pregnancy and found that CD155 expression was higher in the decidua, which was consistent with the aforementioned immunofluorescence results. Additionally, the expression of CD112 in the decidua was also significantly higher than that in the endometrium. However, a literature review revealed that CD112 primarily interacts with CD112R and TIGIT to exert immunosuppressive effects, while single-cell sequencing of the maternal–foetal interface shows that CD112R is scarcely expressed at this site, leading to the limited focus on CD112 here [25]. Based on the above findings, we directed our research towards the functional expression of CD96 in the presence of CD155 at the maternal–foetal interface.

Trophoblast cells share many characteristics with tumour cells in inducing and maintaining immune tolerance, especially in inducing immune tolerance in NK cells [26]. There are a considerable number of dNK cells in the decidua, accounting for approximately 70% of the total number of deciduous lymphocytes [27]. During pregnancy, the decidua maintains close contact with trophoblasts without causing damage because trophoblasts exert an immunosuppressive effect on dNK cells [28]. NK cells in the decidua are a dominant subpopulation, and various factors with inhibitory effects on NK cells are locally present. It has been confirmed that trophoblast cells are resistant to the killing effect of NK cells, and in vitro experiments have shown that purified decidual NK cells fail to produce a cytotoxic response against freshly isolated trophoblast cells. After activation by interleukin-2/IL-2, certain cytotoxic activity against choriocarcinoma cells can be obtained, but trophoblast cells with activated decidual NK cells still retain some degree of resistance [29]. Our experiments also showed that under the conditions of trophoblast cell coculture with NK cells, the cytokine secretion function of NK cells was significantly inhibited, and the expression of cytokines and adhesion molecules was significantly altered. In contrast, the expression of the immune tolerance molecule IL-10 was increased in the coculture.

In this study, we found abnormalities in the expression of CD96 at the pathological maternal–foetal interface of pregnancy, and we also found that CD96 plays an important role in immunosuppression and inducing immune tolerance on the surface of NK cells. In the experiments in this chapter, after we blocked the action of trophoblasts and decidual stromal cells with CD96 using CD96-specific antagonists, the adhesion of dNK cells decreased significantly, and the expression of the corresponding tolerant cytokine IL-10 was also significantly reduced, while the expression of killer cytokines increased, and the proliferative activity of NK cells showed a substantial increase. This change marks the transformation of dNK cells from a robust immune-tolerant phenotype to an active immunoaggressive phenotype, thus demonstrating the substantial impact of CD96 expression on the functional status of dNK cells. If the expression of CD96 in NK cells can be increased in some way to inhibit the function of dNK cells, it could potentially have a beneficial impact on maintaining pregnancy.

Although decidual NK cells exhibit low activity, they can still exert cytotoxic effects on trophoblast cells under certain conditions [18]. In the early stage of embryonic development, decidual NK cells can regulate the growth of the placenta via direct anti-placental trophoblast cytotoxic activity, and as mentioned earlier, spontaneous abortion is also associated with excessive cytotoxic activity [30]. Studies have shown that the cytotoxic activity of IL-2-activated decidual dNK cells against trophoblast cell tumours surpasses that against normal trophoblast cells. This observation suggests that decidual dNK cells likely contribute to regulating trophoblast cell invasion [31,32]. It has also been reported that trophoblast cells interact with dNK cells, resulting in the synthesis and secretion of certain growth factors that stimulate trophoblast cell proliferation, indicating that the cytotoxic activity of dNK cells is widely inhibited during successful pregnancies [33].

An abnormal dNK cell ratio or function is closely related to the occurrence of recurrent pregnancy loss [34]. The cause of the other 50% of cases is unknown, and this condition is called unexplained RPL [35]. These unexplained cases are often associated with immune disorders. Multiple studies have proposed a potential connection between the abnormal number and subpopulation of NK cells, which may relate to uRPL. Most studies have shown that NK cell concentrations are higher in women with uRPL than in healthy fertile women. In addition, NK cells and trophoblast cells interfere with each other. Trophoblast cells can transmit signals to dNK cells through direct contact or regulate the function of NK cells by expressing and secreting a variety of cytokines and chemokines. IL-8, secreted by dNK cells, has been reported to play an important role in placental formation and trophoblast cell invasion [26,36]. Overall, the abnormal interaction between dNK cells and trophoblast cells exhibited a strong correlation with pregnancy failure and miscarriage. Previous studies and the above experiments showed that the enrichment of NK cells during pregnancy will disrupt the homeostasis of pregnancy, and, of course, the intervention of external factors can significantly affect this process [37,38].

Interventional factors can be useful as targets for clinical treatment. Rapamycin, for example, has been shown to mediate autophagy in immune cells during pregnancy while maintaining pregnancy homeostasis [21,27]. Previous studies have shown that palmitic acid exerts a proinflammatory effect at the maternal–foetal interface, which partially explains why adverse pregnancy events are high in obese patients. Studies have also found that fatty acids accumulate early in the maternal–foetal interface [19,39]. Therefore, we studied the role of palmitic acid in the maternal–foetal interface. Palmitic acid, in this study, played a special role, and we found that palmitic acid can affect the oxidative activity of NK cells and inhibit the activity by downregulating the secretion of toxic cytokines. However, the expression of the immune tolerance molecule IL 10, as well as the adhesion molecules ICAM-1, VCAM, and SELE, was significantly increased. This may mean that a low dose of palmitic acid has the potential to keep early pregnancies in a stable state.

Our previous study showed that the adhesion function of NK cells during pregnancy is highly correlated with pregnancy status. In the normal endometrium, the activity function of NK cells is significantly increased, thus mediating immune rejection to protect the endometrium against external harmful factors. At this time, NK cell adhesion is poor, and motility is increased. However, after pregnancy, NK cells transform to a dNK phenotype with a low cytokine secretion function, thus ensuring the stable implantation and development of embryonic tissues containing heterologous genes. Moreover, due to the increased adhesion molecules of NK cells, such as ICAM-1, ICAM-2, VCAM, SELE, SELL, and SELP, NK cells remain more active locally and play a role in maintaining immune homeostasis.

In summary, as shown in Figure 6, maternal–foetal interface trophoblasts can reduce and regulate decidual NK cells to promote the secretion of cytokines such as IFN-γ and granzyme B; upregulate the expression of the inhibitory cytokine IL-10; upregulate the expression of adhesion factors such as CD54, CD106, and CD62E; mediate the in situ adhesion of decidual NK cells; and exert an inhibitory effect on the immunotoxicity of dNK cells. The value of the results of this study for clinical application needs to be further explored in future research.

## Figures and Tables

**Figure 1 bioengineering-10-01008-f001:**
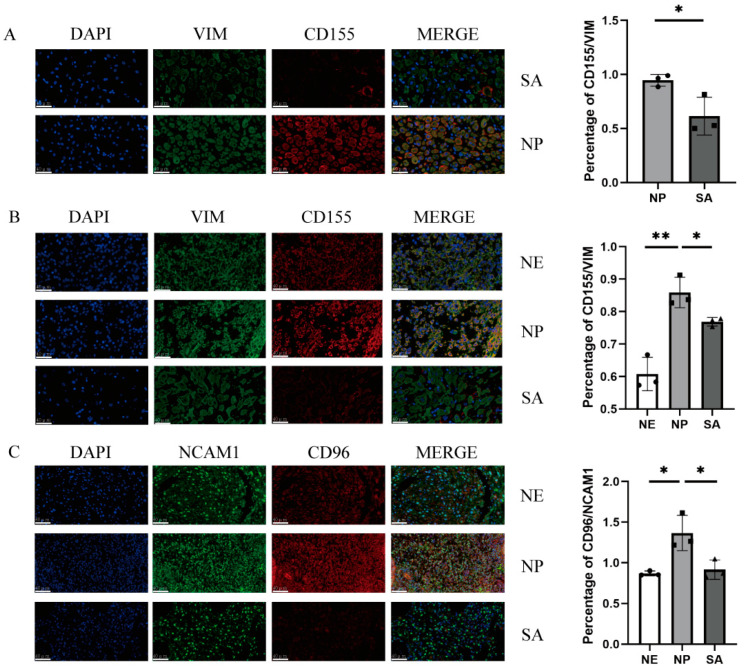
Immunofluorescence staining of the maternal–foetal interface: (**A**) Immunofluorescence staining of villi in normal pregnancy and spontaneous abortion and vimentin-labelled chorionic stromal cells. (**B**) Immunofluorescence staining of normal endometrium, decidua of normal pregnancy, and decidua of spontaneous abortion and vimentin-labelled DSCs or ESCs. (**C**) Immunofluorescence staining of normal endometrium, decidua of normal pregnancy, and decidua of spontaneous abortion and NCAM1-labelled dNK cells. All fields of view were shot under 400× lenses. VIM: vimentin; SA: spontaneous abortion; NE: normal endometrium; NP: normal pregnancy. *: *p* < 0.05; **: *p* < 0.01.

**Figure 2 bioengineering-10-01008-f002:**
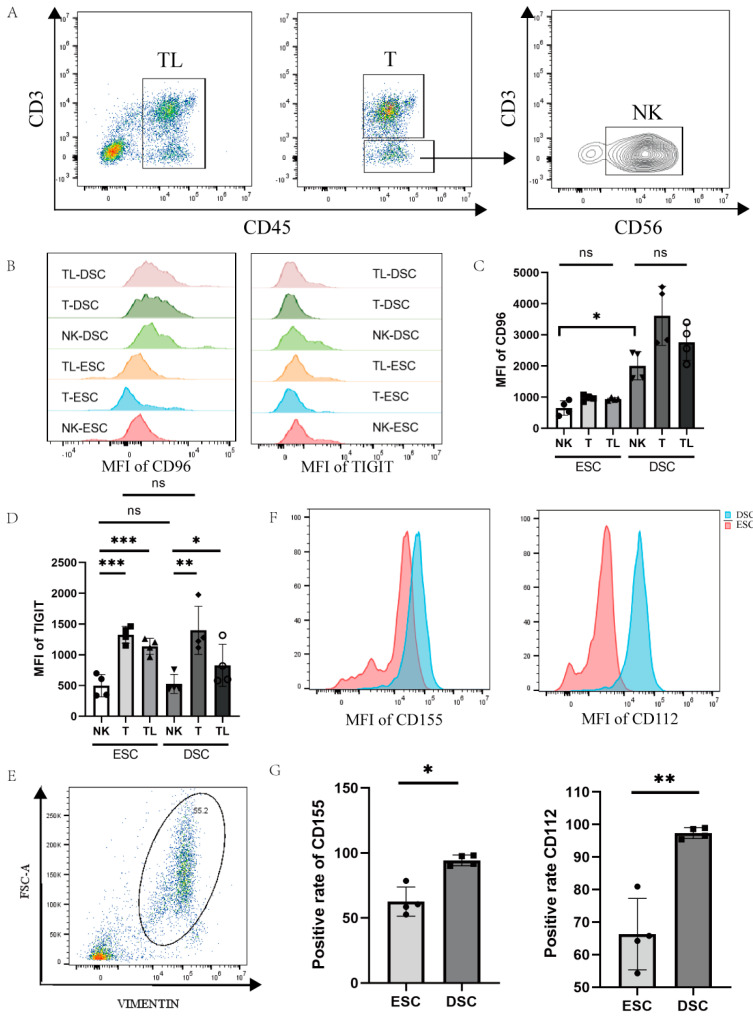
Expression of CD96 and CD155 at the decidual interface in normal pregnancy: (**A**) Gating strategies for NK cells, T cells, and total immune cells. (**B**) Histogram of differences in the expression of CD96 and TIGIT in different cell subsets in decidual and endometrial tissues of normal pregnancy. (**C**) Differences in the average fluorescence intensity of CD96 expression in different subpopulations of cells. (**D**) Differences in the mean fluorescence intensity of TIGIT expression in different cell subpopulations. (**E**) Entrapment gating strategies for stromal cells in deciduous tissue and endometrial tissue. (**F**) Histogram of differences in the expression of CD155 and CD112 in DSCs and ESCs. (**G**) Statistics of the difference in the expression of CD155 and CD112 in DSCs and ESCs. DSCs: deciduous stromal cells; ESCs: endometrial stromal cells. TL: total lymphocyte; DSC: decidual stromal cell; ESC: endometrial stromal cell; MFI: mean fluorescence intensity. ns: *p* > 0.05; *: *p* < 0.05; **: *p* < 0.01; ***: *p* < 0.001.

**Figure 3 bioengineering-10-01008-f003:**
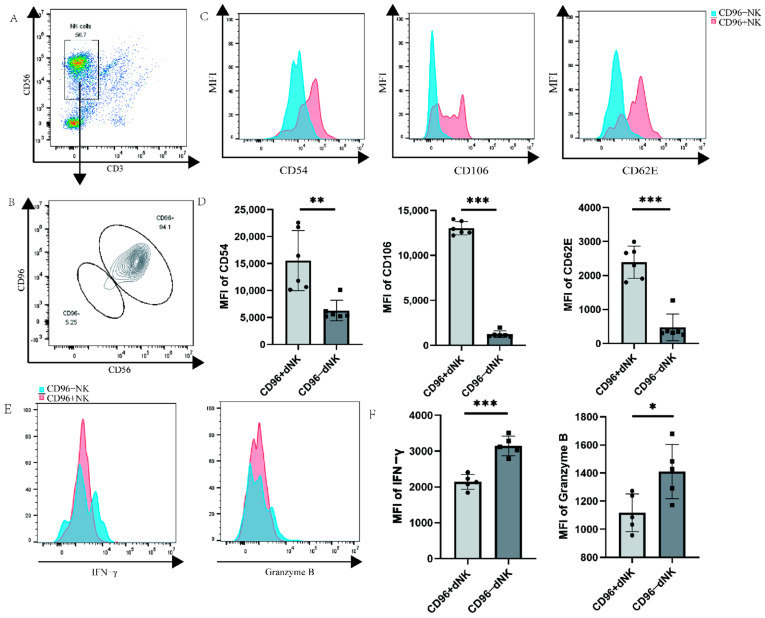
Functional molecule expression on CD96+ dNK cells and CD96− dNK cells: (**A**) The population and proportion of NK cells in primary cells. (**B**) The expression of CD96 on NK cells circled in (**A**). (**C**) Histogram of the expression intensity of the three adhesion factors: CD54, CD62E, and CD106, as tested via flow cytometry. (**D**) The mean fluorescence intensity of the three adhesion factors was measured via flow cytometry, and the differences were calculated. (**E**) Histogram of the expression intensity of IFN-γ and granzyme B as measured via flow cytometry. (**F**) The mean fluorescence intensity of the two cytokines was measured via flow cytometry, and the difference was calculated. MFI: mean fluorescence intensity. *: *p* < 0.05; **: *p* < 0.01; ***: *p* < 0.001.

**Figure 4 bioengineering-10-01008-f004:**
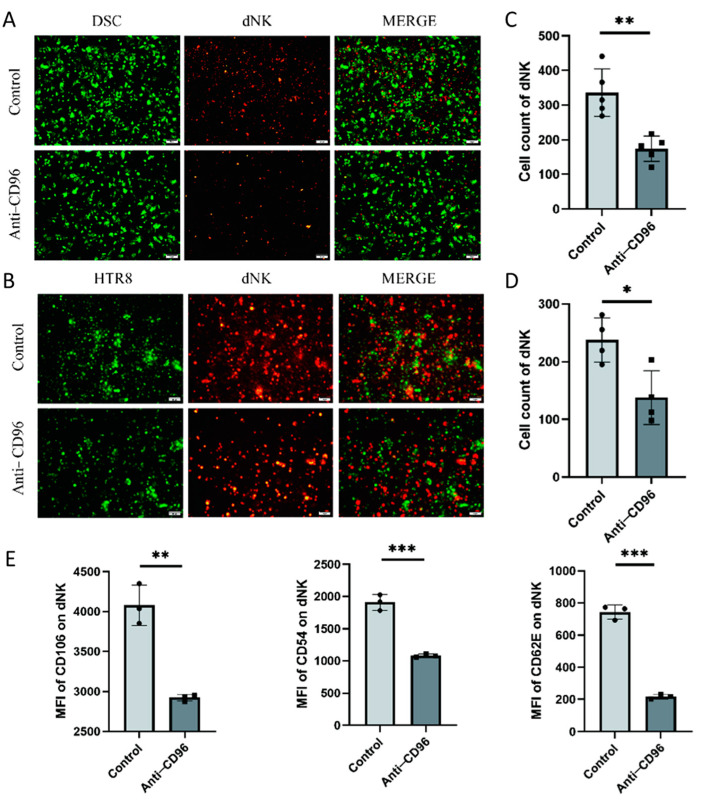
After the addition of CD96 antagonists, the function of NK cells in the coculture system decreased significantly. (**A**) Cell fluorescence staining assay to detect the adhesion of NK cells to DSCs before and after the addition of antagonists (400× magnification). (**B**) Cell fluorescence staining to detect the adhesion of NK cells to HTR8 cells before and after the addition of antagonists (400× magnification). (**C**) Number of dNK cells adhered to the surface of DSCs. (**D**) Number of dNK cells adhered to the surface of HTR8. (**E**) Flow cytometry to detect the effect of adding antagonists on the expression of NK cell adhesion molecules. (**F**) Flow cytometry to detect the expression intensity histogram of three cytokines. (**G**) Flow cytometry was used to detect the average fluorescence intensity of the three cytokines and count their differences. MFI: mean fluorescence intensity. *: *p* < 0.05; **: *p* < 0.01; ***: *p* < 0.001.

**Figure 5 bioengineering-10-01008-f005:**
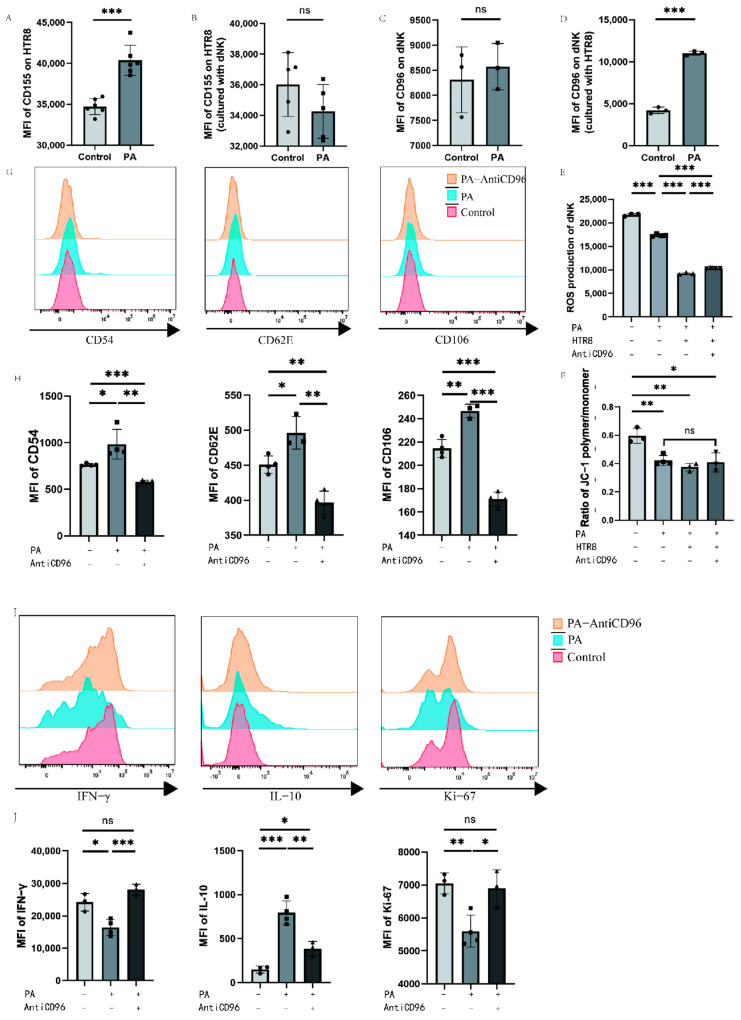
Palmitic acid may inhibit NK cell activity through CD96 in dNK cells. (**A**) Effect of palmitic acid on CD155 expression in HTR8 T cells. (**B**) Effect of palmitic acid on CD155 expression in HTR8 cells cocultured with dNK cells. (**C**) Effect of palmitic acid on CD96 expression in dNK cells. (**D**) Effect of palmitic acid on CD96 expression in dNK cells cocultured with HTR8 cells. (**E**) ROS content of dNK cells under different conditions. (**F**) Ratio of JC-1 polymer/monomer of dNK cells under different conditions. (**G**) Histogram of the expression intensity of the three adhesion molecules detected via flow cytometry. (**H**) The mean fluorescence intensity of the three adhesion molecules was measured via flow cytometry, and the differences were calculated. (**I**) Histogram of the expression intensity of the three adhesion cytokines detected via flow cytometry. (**J**) The mean fluorescence intensity of the three cytokines were measured via flow cytometry, and the differences were calculated. PA: palmitic acid; MFI: mean fluorescence intensity. ns: *p* > 0.05; *: *p* < 0.05; **: *p* < 0.01; ***: *p* < 0.001.

**Figure 6 bioengineering-10-01008-f006:**
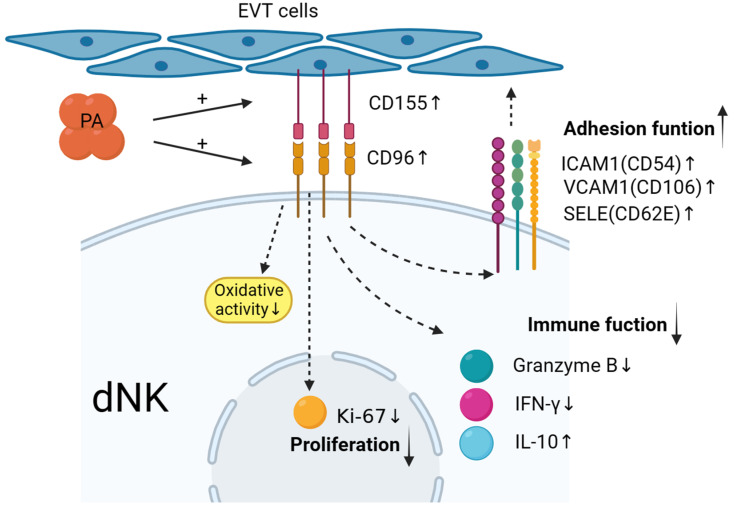
Low-dose palmitic acid upregulates the surface expression of CD155 on trophoblast cells and CD96 on decidual natural killer (dNK) cells, resulting in decreased expression of cytotoxic cytokines in dNK cells. Moreover, it leads to an increase in inhibitory cytokine IL-10 and adhesion molecules ICAM1, VCAM1, and SELE. These changes signify the transition of dNK cell function towards an immune-tolerant phenotype. Additionally, there is a significant decrease in the expression of the proliferation marker Ki-67 in dNK cells. These findings suggest the potential involvement of these factors in the regulation of early-pregnancy maternal–foetal immune tolerance.

**Table 1 bioengineering-10-01008-t001:** Flowcytometry antibodies used in this article.

Antibody	Fluorescence	Manufactory	Clone
Anti-human CD45 antibody	BV510	Biolegend	2D1
Anti-human CD3 antibody	AF700	Biolegend	SK7
Anti-human CD56 antibody	PE/Cy7	Biolegend	HCD56
Anti-human TIGIT antibody	PE	Biolegend	A15153G
Anti-human CD96 antibody	BV421	Biolegend	NK92.39
Anti human CD96 antibody	AO	Abcam	NK92.39
Anti human CD155 antibody	PE	Biolegend	TX24
Anti human CD112 antibody	APC	Biolegend	TX31
Anti-human Vimentin antibody	AF488	BD	RV202
Anti-human Granzyme B antibody	APC	Biolegend	QA18A28
Anti-human CD54 antibody	FITC	Biolegend	HA58
Anti-human CD62E antibody	PE	Biolegend	HAE-1f
Anti-human CD106 antibody	APC	Biolegend	STA
Anti-human IL10 antibody	APC	Biolegend	JES3-9D7
Anti-human Ki67 antibody	APC	Biolegend	Ki-67
Anti-human Ki67 antibody	FITC	Biolegend	11F6
Anti-human IFN-γ antibody	AF700	Biolegend	4S.B3
Anti-human IFN-γ antibody	BV421	Biolegend	4S.B3

## Data Availability

The data presented in this study are available on request from the corresponding author. The data are not publicly available due to privacy or ethical reasons.

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
