# Peer review of "Palmitic Acid Upregulates CD96 Expression to Mediate Maternal–Foetal Interface Immune Tolerance by Inhibiting Cytotoxic Activity and Promoting Adhesion Function in Human Decidual Natural Killer Cells"

_bioengineering, 2023, doi:10.3390/bioengineering10091008_

Round 1

Reviewer 1 Report

In this paper, the mechanism of palmitic acid to mediate maternal-foetal interface immune tolerance through CD96 upregulation has been investigated. It showed that higher CD96 expression could inhibit the cytotoxic activity and promote cellular adhesion of human decidual NK cells. Overall the manuscript is clearly demonstrate the scientific merits of the proposed work, and thus it is likely acceptable to be published after subsequent minor issues are addressed properly. Here are some minor comments:

1. How was the 10 ug/ml concentration of palmitic acid being determined in this experiment? 

2. To the best of my knowledge, cell line would undergo a phenotypic changes as compared to primary cells. Thus, is there any significance issue of using trophoblast cell line (HTR8s) for coculturing with primary dNK cells by any chance?

3. Please clearly define all used abbreviations in the main text, not only in the figure legend.

4. In section 2.5, it seems that materials and methods in this whole paragraph is numbered. It might be better to remove the number among the sentences as it appears strange.

5. In section 2.7, please fix the formatting for number of cells, as it should be in superscript format.

6. The introduction and discussion seem to have an error in citing references, as shown by odds number appeared at the end of some sentences. Please check and fix it.

Author Response

  1. How was the 10 ug/ml concentration of palmitic acid being determined in this experiment? 

Response:In Chapter 3.5 of this manuscript, the use of palmitic acid was mentioned at a concentration of 10μM, rather than 10 μg/ml. In the literature concerning palmitic acid, cellular treatment concentrations typically exceed 100μM, which has been shown to exhibit pro-inflammatory effects at the maternal-fetal interface and lead to adverse pregnancy outcomes. However, in Figure 7 of the article "Single-cell transcriptome profiling of the human endometrium of patients with recurrent implantation failure" (PMID: 36185612), it was observed that a low concentration of 10μM palmitic acid upregulated the expression of CXCL12 and IL15 at the maternal-fetal interface, potentially contributing to the prevention of recurrent implantation failure. To further investigate the positive effects of low-dose palmitic acid at the maternal-fetal interface, the present study utilized a concentration of 10μM palmitic acid for cellular treatment. This article has been cited and supplemented with relevant information in the introduction and Chapter 3.5.

  1. To the best of my knowledge, cell line would undergo a phenotypic changes as compared to primary cells. Thus, is there any significance issue of using trophoblast cell line (HTR8s) for coculturing with primary dNK cells by any chance?

Response:CD155, located on the surface of trophoblast, is the main pathway involved in the negative regulation of CD96 on dNK cell surfaces. Therefore, the effects of CD96 antagonists can only be observed under conditions where dNK cells are co-cultured with nourishing cells. The main observation targets in this study is the functional characteristics of dNK cells. To eliminate the uncertainty caused by individual differences in primary trophoblast, the trophoblast cell line HTR8s was used for co-culture with dNK cells.

  1. Please clearly define all used abbreviations in the main text, not only in the figure legend.

Response:Thanks for your reminding. All abbreviations used in the text have been clearly defined upon their first occurrence.

  1. In section 2.5, it seems that materials and methods in this whole paragraph is numbered. It might be better to remove the number among the sentences as it appears strange.

Response:Thanks for your reminding. The numbering in the Materials and Methods paragraph has been removed and revised to a smoother paragraph.

  1. In section 2.7, please fix the formatting for number of cells, as it should be in superscript format.

Response:Thanks for your reminding. The cell count format has been corrected to superscript.

  1. The introduction and discussion seem to have an error in citing references, as shown by odds number appeared at the end of some sentences. Please check and fix it.

Response:Thanks for your reminding. The literature citation format appears to have been incorrect in the format conversion of the document and has been corrected in the revised version of the manuscript.

Finally, thank you again for your guidance and for reviewing and revising my paper again. I hope that under your guidance, I can complete this excellent paper, and sincerely hope that my paper can be published in your journal.

Reviewer 2 Report

This study by Wang et al reports expression of CD96 family members at the maternal fetal interface during early pregnancy. This is a novel study as although dNK cells are demonstrated to play an important role in pregnancy implantation and recurrent pregnancy loss, the role of CD96 family receptors on dNK cells has previously only limited investigation at the maternal fetal interface. This study demonstrates increased expression of CD96 on dNK cells from normal pregnancy and reduced CD96 expression in samples from patients with spontaneous abortion. They also demonstrate that inhibition of CD96 on dNK cells results in increased cytotoxicity, while administration of palmitic acid upregulates expression of CD96 and reduces cytotoxic ability. These data are interesting findings which advance our knowledge of dNK cell function in pregnancy, however, several experimental issues need to be addressed prior to publication. 

Major points:

1. The materials and methods section is inadequate to describe the experiments performed. In section 2.2, the specific assays used for cytokine and protein detection should be described. In section 2.4, the flow cytometry antibodies used should be stated by catalogue number. It is unclear in each experiment whether the dNK cells evaluated by flow cytometry are from fresh tissue or cell culture. It is unclear whether the flow cytometry protocols evaluated for cell viability which would have a significant effect on results if unaccounted for. The number of samples used in each assay are not mentioned. 

2. In figure 3 E-F, the protocol for intracellular staining is not described. Typically intracellular IFNg and granzyme B levels are assessed after NK cell stimulation, was this performed? The differences in IFNg and GZMB MFI reported in 3F are not consistent with the figure 3E flow data.

3. The rationale for testing palmitic acid on dNK cells is unclear. In the introduction the authors state "We suspect that palmitic acid may be beneficial for pregnancy maintenance at this time, but no studies have yet confirmed this conjecture." The reason for assessing palmitic acid should be clarified and references added to support this conjecture.

4. The authors state that palmitic acid may lead to increased apoptosis in dNK cells, however this is supported by a single apoptosis assay in Figure 5F and the data demonstrate that this is not affected by CD96 inhibitors. Additional experiments should be performed to demonstrate apoptosis related to CD96 function or these conclusions should be removed as they are not supported by current data.

5. The authors conclude that "The high expression of CD96 in spontaneous abortion patients indicates the enhanced killing activity of NK cells, which may be one of the possible factors of pregnancy failure." However their data in figure 1C demonstrate lower CD96 expression in spontaneous abortion patients compared to normal pregnancy and their experimental data demonstrates CD96-related dNK cell suppression. This should be clarified.

6. The authors report several times in the discussion that CD96 inhibited dNK cell killing, however, they do not demonstrate any cytotoxicity assays or other assessment of NK cell killing in this manuscript so these assays should be performed or this conclusion should be removed.

7. A recent study by Habets et al. (PMID: 36004818) found increased expression of CD96 on PBMCs from recurrent pregnancy loss samples in contrast to the current study which found decreased CD96 on dNK cells. The difference in these findings from the current manuscript should be added to the discussion and this paper should be cited. 

Minor points:

1. In the introduction (page 2, line 67) the authors state "PVR is also a high-affinity ligand for CD96, a receptor that inhibits NK and T-cell cytotoxicity. In addition, PVR binds to CD96 (tactile) receptors and has activation and inhibitory functions on NK cells." These sentences are redundant and should be edited. 

2. Ki-67 is repeatedly referred to as a cytokine, which is incorrect. Rather it is a marker of cell proliferation.

Overall the English language use is appropriate, however there are several  grammatical errors and sentence structures than would benefit from moderate editing. 

Author Response

Reviewer2

Major points:

  1. The materials and methods section is inadequate to describe the experiments performed. In section 2.2, the specific assays used for cytokine and protein detection should be described. In section 2.4, the flow cytometry antibodies used should be stated by catalogue number. It is unclear in each experiment whether the dNK cells evaluated by flow cytometry are from fresh tissue or cell culture. It is unclear whether the flow cytometry protocols evaluated for cell viability which would have a significant effect on results if unaccounted for. The number of samples used in each assay are not mentioned. 

Response:Thanks for your reminding.

Flow cytometry has been used for cytokine and protein detection in this paper, as we have described in Sections 2.2 and 2.4 for additional information.

The flow cytometry antibodies used in this paper have been listed in a separate Table1.

The dNK cells for each part of the experiment were obtained from fresh primary tissue isolation and were cultured and treated accordingly.

In this study, primary cells were isolated and used for various experiments. The success of primary cell isolation was determined by achieving a cell concentration greater than 1×106 cells/ml. Decidual stromal cells and endometrial stromal cells were considered active if they adhered to the cell culture plates within 2 hours after isolation. The primary immune cells were considered active and pure if the percentage of CD45 positivity exceeded 90%. CD56 was used as a marker to identify primary NK cells isolated from the primary immune cell population.All primary cells were cultured in well plates and subjected to different treatments within a 24-hour timeframe. Flow cytometry staining was performed within 48 hours of culture to assess cell phenotypes. This ensured that the primary cells were maintained in optimal conditions during the experiments.To ensure accurate flow cytometry analysis, the flow cytometry protocol excluded the lower left corner area within the FSC-A/SSC-A gate, which typically represents cellular debris. Additionally, cells falling on the diagonal within the FSC-A/FSC-H gate were circled to exclude cell clusters. These measures ensured that only single live cells were included in the flow cytometry analysis.

All statistical graphs in the Figures have been redesigned to present the sample sizes for each experiment in the form of scatter plots.

  1. In figure 3 E-F, the protocol for intracellular staining is not described. Typically intracellular IFNg and granzyme B levels are assessed after NK cell stimulation, was this performed? The differences in IFNg and GZMB MFI reported in 3F are not consistent with the figure 3E flow data.

Response:Thanks for your reminding. We performed cell stimulation prior to intracellular flow cytometry staining, and the reagents and experimental protocols used are described in the supplemental section 2.4.

In Figure 3E, the main peak of the histogram for CD96+ is more left-skewed than the histogram for CD96- although the distribution pattern does not look very obvious, and the statistics in Figure 3F are calculated based on this.

  1. The rationale for testing palmitic acid on dNK cells is unclear. In the introduction the authors state "We suspect that palmitic acid may be beneficial for pregnancy maintenance at this time, but no studies have yet confirmed this conjecture." The reason for assessing palmitic acid should be clarified and references added to support this conjecture.

Response:Thanks for your reminding. Relevant content and reference have been added in the corresponding section in INTRODUCTION.

  1. The authors state that palmitic acid may lead to increased apoptosis in dNK cells, however this is supported by a single apoptosis assay in Figure 5F and the data demonstrate that this is not affected by CD96 inhibitors. Additional experiments should be performed to demonstrate apoptosis related to CD96 function or these conclusions should be removed as they are not supported by current data.

Response:Thanks for your reminding.The conclusions of the JC1 experiment regarding apoptosis have been corrected in the text.

  1. The authors conclude that "The high expression of CD96 in spontaneous abortion patients indicates the enhanced killing activity of NK cells, which may be one of the possible factors of pregnancy failure." However their data in figure 1C demonstrate lower CD96 expression in spontaneous abortion patients compared to normal pregnancy and their experimental data demonstrates CD96-related dNK cell suppression. This should be clarified.

Response:Thanks for your reminding. The sentence mentioned is revised in the first paragraph of the discussion section.

  1. The authors report several times in the discussion that CD96 inhibited dNK cell killing, however, they do not demonstrate any cytotoxicity assays or other assessment of NK cell killing in this manuscript so these assays should be performed or this conclusion should be removed.

Response:Thanks for your reminding. The NK cell killing function mentioned in this paper is indirectly reflected by detecting the expression of cytokines related to NK cell activity, such as IFN-γ and granzyme B. In order to make the language of the article more precise, we have accepted your suggestion and revised the corresponding part.

  1. A recent study by Habets et al. (PMID: 36004818) found increased expression of CD96 on PBMCs from recurrent pregnancy loss samples in contrast to the current study which found decreased CD96 on dNK cells. The difference in these findings from the current manuscript should be added to the discussion and this paper should be cited. 

Response: Thanks for your reminding. The article you mentioned(PMID: 36004818) is very helpful in enriching the discussion of this paper and our understanding of the mechanisms involved, and we have cited it in the discussion section.

Minor points:

  1. In the introduction (page 2, line 67) the authors state "PVR is also a high-affinity ligand for CD96, a receptor that inhibits NK and T-cell cytotoxicity. In addition, PVR binds to CD96 (tactile) receptors and has activation and inhibitory functions on NK cells." These sentences are redundant and should be edited. 

Response:Thanks for your reminding. The sentences have been edited more simple and concise.

  1. Ki-67 is repeatedly referred to as a cytokine, which is incorrect. Rather it is a marker of cell proliferation.

Response:Thanks for your reminding. The expression related to Ki-67 was corrected to cell proliferation marker.

Comments on the Quality of English Language

Overall the English language use is appropriate, however there are several grammatical errors and sentence structures than would benefit from moderate editing. 

Response:Thanks for your reminding. Grammar and sentence structure have been re-proofread and corrected.

Finally, thank you again for your guidance and for reviewing and revising my paper again. I hope that under your guidance, I can complete this excellent paper, and sincerely hope that my paper can be published in your journal.

Reviewer 3 Report

The manuscript by Wang et al. investigated the immunomodulatory role of CD96 in decidual NK cells and its function at the maternal-fetal interface. Below are my comments:

Major points

1. The population of CD96- dNK cells shown in this study is questionable. The expression of CD56 appears to have a close relationship with that of CD96 (Fig. 3B). However, the current gating strategy for the NK cells based on CD56 is too broad (Fig. 3A) and could include other cell types. If a more stringent gating approach for CD56 is employed, it raises the question of whether there remains a CD96-negative population given the small percentage of the CD96- population.

2. Important controls are missing for some experiments:

290-293: the meaning of this experiment is not clear, and the current  conclusion is not related to function of CD96. A comparison between with ot without CD96 antagonists should be performed.

320-321: the group showing coculture with HTR8 but without PA is missing.

Also, comparison should be performed under the same condition; the current data cannot support the conclusion.

3. There are quite some confusing and incorrect statements throughout the manuscript (see below). Some statements even have opposite meaning to the previous sentences. The authors need to provide more accurate description or writing to improve the clarity of the manuscript.

Minor points

16-20: there are obviously more than four parts based on the description.

59: more introduction on CD56 is needed.

64: the description is incorrect. See PMID: 32043568.

67-69: the meaning here is confusing; these two sentences conflict with each other.

77-81: the statement is confusing and should be rewritten.

193-194: “higher” compared to what?

194-195: “lower” compared to what?

208-211: the meaning is confusing.please describe clearly which markers are used for which cell type.

213-214: does this sentence mean the same thing as the previous sentence?

216-217: confusing, as nk cells also belong to immunocytes.

218-220: conflict with the previous sentence.

225-226: “higher” compared to what?

276: incorrect unit for concentration.

288: ki67 is not a cytokine.

307-314: it is not clear what conclusion the authors are trying to draw here.

319-320: results not shown for the statement.

327-329: as the authors mentioned that there is no statisitc difference in the previous sentence, how can they draw this conclusion?

331 (Fig. 5F): The effect of HTR8 cells is weak, and there may be doubt regarding whether there is statistical difference between “PA” and “PA-HTR8”.

338-340: from Fig. 5G, there is no difference among the three groups.

Fig. 5: description for “control” is missing for all the panels.

492-494: what does “XXX” mean here?

Editing of English language is required.

Author Response

Reviewer3

Major points

  1. The population of CD96- dNK cells shown in this study is questionable. The expression of CD56 appears to have a close relationship with that of CD96 (Fig. 3B). However, the current gating strategy for the NK cells based on CD56 is too broad (Fig. 3A) and could include other cell types. If a more stringent gating approach for CD56 is employed, it raises the question of whether there remains a CD96-negative population given the small percentage of the CD96- population.

Response:Thanks for your reminding.Several articles are cited in the discussion section of this paper discussing CD96 as a surface marker expressed on T cells and NK cells. It is well known that CD3+ is a characteristic marker for T cells and CD56+ is a characteristic marker for NK cells, so the close relationship between CD96 and CD56 should be understood and accepted when we have excluded T cells and circled NK cells using a CD3-CD56+ flow cytometry gating strategy.

  1. Important controls are missing for some experiments:

290-293: the meaning of this experiment is not clear, and the current  conclusion is not related to function of CD96. A comparison between with or without CD96 antagonists should be performed.

Response:Thanks for your reminding. The experiment was performed with dNK cells and HTR8 co-culture as control group and the addition of CD96 antagonist as experimental group, with the intention to investigate whether CD96 molecules on dNK cells act directly between the trophoblast cell line HTR8 and dNK cells. The difference between the control and experimental groups was whether or not the CD96 antagonist was added to the culture system.

320-321: the group showing coculture with HTR8 but without PA is missing.

Also, comparison should be performed under the same condition; the current data cannot support the conclusion.

Response:Thanks for your reminding. Since the main goal was to observe the effect of PA on ROS levels in dNK cells under different conditions, a dNK and HTR8 co-culture group was not set up. The conclusion of this experiment was drawn by comparing adjacent groups in Fig. 5 E, and all the adjacent groups involved only one variable change between them, thus achieving the control of conditions.

  1. There are quite some confusing and incorrect statements throughout the manuscript (see below). Some statements even have opposite meaning to the previous sentences. The authors need to provide more accurate description or writing to improve the clarity of the manuscript.

Response:Thanks for your reminding. The statements have been revised as mentioned in Minor points.

Minor points

16-20: there are obviously more than four parts based on the description.

Response: Thanks for your reminding. The expression has been modified.

59: more introduction on CD56 is needed.

Response: Thanks for your reminding. Further additions have been made to the introduction on CD56.

64: the description is incorrect. See PMID: 32043568.

Response: Thanks for your reminding. The expression has been revised,and the article mentioned has been cited.

67-69: the meaning here is confusing; these two sentences conflict with each other.

Response: Thanks for your reminding. The expression has been revised.

77-81: the statement is confusing and should be rewritten.

Response: Thanks for your reminding. The expression has been rewritten.

193-194: “higher” compared to what?

Response: Thanks for your reminding. The expression has been revised.

194-195: “lower” compared to what?

Response: Thanks for your reminding. The expression has been revised.

208-211: the meaning is confusing.please describe clearly which markers are used for which cell type.

Response: Thanks for your reminding. The expression has been revised.

213-214: does this sentence mean the same thing as the previous sentence?

Response: Thanks for your reminding. The sentence is semantically repetitive with the previous sentence and has been deleted.

216-217: confusing, as nk cells also belong to immunocytes.

Response: Thanks for your reminding. Total lymphocytes include T-cell NK cells as well as other cells not indicated, and the intention here is to compare the phenotypic differences between NK cells, T cells and lymphocytes as an overall concept.

218-220: conflict with the previous sentence.

Response: Thanks for your reminding. The expression has been revised.

225-226: “higher” compared to what?

Response: Thanks for your reminding. The expression has been revised.

276: incorrect unit for concentration.

Response: Thanks for your reminding. A correction has been made to the concentration unit.

288: ki67 is not a cytokine.

Response: Thanks for your reminding. The expression related to Ki-67 was corrected to cell proliferation marker.

307-314: it is not clear what conclusion the authors are trying to draw here.

Response: Thanks for your reminding. The experimental design and description of the results in this section are well described.

319-320: results not shown for the statement.

Response: Thanks for your reminding. The expression has been revised.

327-329: as the authors mentioned that there is no statisitc difference in the previous sentence, how can they draw this conclusion?

Response:Thanks for your reminding. The results of this experiment did not show statistical differences. The conclusions of the JC1 experiment regarding apoptosis have been corrected in the text.

331 (Fig. 5F): The effect of HTR8 cells is weak, and there may be doubt regarding whether there is statistical difference between “PA” and “PA-HTR8”.

Response:Thanks for your reminding. The results of this experiment did not show statistical differences. The conclusions of the JC1 experiment regarding apoptosis have been corrected in the text.

338-340: from Fig. 5G, there is no difference among the three groups.

Response:Thanks for your reminding. Although the peak positions of the histograms were not significantly shifted, the mean fluorescence intensities showed statistical differences upon counting.

Fig. 5: description for “control” is missing for all the panels.

Response: Thanks for your reminding. The control group settings for this section are described in detail in the text.

492-494: what does “XXX” mean here?

Response:The content of this paragraph is the Institutional Review Board Statement, which has been supplemented by the actual situation.

Comments on the Quality of English Language

Editing of English language is required.

Response: Thanks for your reminding. The language grammar has been re-proofread and corrected.

Finally, thank you again for your guidance and for reviewing and revising my paper again. I hope that under your guidance, I can complete this excellent paper, and sincerely hope that my paper can be published in your journal.

Round 2

Reviewer 2 Report

The authors have adequately addressed reviewers prior critiques. The manuscript is significantly improved, specifically the clarity of methods and discussion sections and appropriate for publication with very minor changes. 

Minor issue: in table 1, the clone or catalog # of the antibodies should be added for clarify

English language is improved with only minor edits needed.

Author Response

Minor issue: in table 1, the clone or catalog # of the antibodies should be added for clarify

Response:Thanks for your reminding. Clone numbers for all flow cytometry antibodies have been added in Table 1.

Comments on the Quality of English Language

English language is improved with only minor edits needed.

Response:Thanks for your reminding. Corrections have been made to the language of this article.

Finally, thank you again for your guidance and for reviewing and revising my paper again. I hope that under your guidance, I can complete this excellent paper, and sincerely hope that my paper can be published in your journal.

Reviewer 3 Report

Thanks for the authors' response. Below are my comments:

1. The CD96- dNK population is still questionable. Any reasons to include the sparse cells between CD56+ and CD56- populations in the current gate? I would suggest the authors to compare the gating strategy of CD56+ and CD96- in other publications.

2. Why is PMA needed after the co-culture? Did the control group also include the PMA treatment? Also in Line 317 authors mentioned increase of Ki-67 (Fig. 4F), but in Fig. 4F "Ki-67" panel the "control" group actually showed higher signals than the "anti-CD96" group, which is against the conclusion.

Author Response

  1. The CD96- dNK population is still questionable. Any reasons to include the sparse cells between CD56+ and CD56- populations in the current gate? I would suggest the authors to compare the gating strategy of CD56+ and CD96- in other publications.

Response:Thanks for your reminding.

Our NK cell circle-gate strategy is delineated near the middle of the CD56- and CD56+ cell populations on the scatter plot, which is visually evident in the contour plots as well, predominantly appearing within the depression of the cell distribution (ResponseFigure 1B). This "depression" serves as a criterion for distinguishing between negative and positive groups and thus encompasses a few sparsely distributed cells beyond the primary cell clusters. In accordance with your suggestion, we conducted a review of other articles on the circle gate method and selected a representative image for reference (ResponseFigure 1A). Notably, this image, authored by Wang JC (PMID: 31313887), does not include the sparse cells you mentioned during NK cell gating.

In replicating similar experiments within this study, wherein NK cells were circle-gated using the aforementioned approach, we consistently observed that the CD96+ subset accounted for an overwhelming majority of approximately 90% within the CD56+ population of decidual NK cells. Meanwhile, the CD56- population primarily exhibited CD96- expression (ResponseFigure 1C,D). These findings align with our present study's outcomes.

To summarize, despite adopting a different cell gating strategy, the alterations made to the gating methodology did not introduce any significant bias to the experimental results. The conclusions drawn remained consistent, indicating that the vast majority of decidual NK cells exhibited a CD96+ phenotype, with CD96 expression exhibiting a strong correlation with CD56.

ResponseFigure 1. The results of several flow cytometry experiments on CD56 and CD96 circle gates are demonstrated in the literature and conducted by us. (A)NK cell circle-gate strategy in other articles. (B)Further demonstration of the NK cell circle-gate strategy in Figure 3A in the main text. (C)(D)Two examples of CD96 assay that we performed in other experiments in this study.

  1. Why is PMA needed after the co-culture? Did the control group also include the PMA treatment? Also in Line 317 authors mentioned increase of Ki-67 (Fig. 4F), but in Fig. 4F "Ki-67" panel the "control" group actually showed higher signals than the "anti-CD96" group, which is against the conclusion.

Response: Thanks for your reminding.

PMA is one of the components of the Cell Activation Cocktail and is used to stimulate cells to enrich for cytokines and facilitate cytokine detection. This reagent is described in Section 2.4 and it is used in experiments for all cytokine assays. We have modified the description related to PMA to avoid confusion.

In Fig. 4F, the colors of the three histograms of the results of the three cytokine assays were set reversed for the control and experimental groups, and the labeling of Fig. 4F has been recorrected.

Finally, thank you again for your guidance and for reviewing and revising my paper again. I hope that under your guidance, I can complete this excellent paper, and sincerely hope that my paper can be published in your journal.

Round 3

Reviewer 3 Report

Major point 1: 

Yes, the CD56+ gating should only include the major CD56+ population as shown in Response Fig. 1A; however, the current Fig. 3A still includes the sparse cells away from the major CD56+ cells, which is not a appropriate way to charaterize the CD56+ population. If only the major CD56+ cells are gated, I doubt whether the tiny CD96- population can still be observed. Based on this, even though the authors observed some difference between CD96+ and CD96- cells (Fig. 3 D and F), it brings up the concern whether these are CD56+ cells.

Major point 2:

Fig. 5 C and D: the difference caused by co-culture with HTR8 is actually more obvious than that caused by PA, when comparing the "control" groups between C and D (both without PA). Also, the standard deviations in 5C are so much larger than those in 5D, why is that? Fig. 5G, the difference between the three groups are too minor as shown in the flowcytometry plots.

Cell activation cocktail: It seems odd that the addition of cell activation cocktail is needed before the detection these markers; this is not reflecting the actuall functional difference under physiological conditions. Is there any other references to show this is an appropirate way for such measurements?